

# Evolutionary analysis of the OSCA gene family in sunflower (*Helianthus annuus L*) and expression analysis under NaCl stress

Feibiao Shan[1,2,3,*], Yue Wu[3,*], Ruixia Du[3], Qinfang Yang[3], Chunhui Liu[3], Yongxing Wang[3], Chun Zhang[1] and Yang Chen[1,3]

[1] Hetao College, Bayannur, China

[2] Inner Mongolia Traditional Chinese & Mongolian Medical Research Institute, Hohhot, China

[3] Bayannur Institute of Agricultural and Animal Sciences, Bayannur, China

[*] These authors contributed equally to this work.

## ABSTRACT

Hyperosmolality-gated calcium-permeable channels (OSCA) are $Ca^{2+}$ nonselective cation channels that contain the calcium-dependent DUF221 domain, which plays an important role in plant response to stress and growth. However, the OSCA gene has not been fully identified and analyzed in sunflowers. In this study, we comprehensively analyzed the number, structure, collinearity, and phylogeny of the OSCA gene family in the sunflower, six Compositae species (*Arctium lappa, Chrysanthemum morifolium, Cichorium endivia, Cichorium intybus, Lactuca sativa var. Angustata,* and *Carthamus tinctorius*), and six other plants (soybean, *Arabidopsis thaliana*, rice, grape, and maize). The expression of the sunflower OSCA gene in nine different tissues, six different hormones, and NaCl stress conditions were analyzed based on transcriptome data and qRT–PCR. A total of 15 OSCA proteins, distributed on 10 chromosomes, were identified in the sunflower, and all of them were located in the endoplasmic reticulum. Using the phylogenetic tree, collinearity, gene structure, and motif analysis of the six Compositae species and six other plants, we found that the sunflower OSCA protein had only three subfamilies and lacked the Group 4 subfamily, which is conserved in the evolution of Compositae and subject to purification selection. The OSCA gene structure and motif analysis of the sunflower and six Compositae showed that there was a positive correlation between the number of motifs of most genes and the length of the gene, different subfamilies had different motifs, and the Group 4 subfamily had the smallest number of genes and the simplest gene structure. RNA-seq and qRT–PCR analysis showed that the expression levels of most OSCA genes in the sunflower changed to varying degrees under salt stress, and *HaOSCA2.6* and *HaOSCA3.1* were the most important in the sunflower's response to salt stress. The coexpression network of the sunflower genes under salt stress was constructed based on weighted gene co-expression network analysis (WGCNA). In conclusion, our findings suggest that the OSCA gene family is conserved during the sunflower's evolution and plays an important role in salt tolerance. These results will deepen our understanding of the evolutionary relationship of the sunflower OSCA gene family and provide a basis for their functional studies under salt stress.

Corresponding authors
Yang Chen, chenyangrz@126.com
Chun Zhang,
zhangchun6625@163.com

## INTRODUCTION

The sunflower (*Helianthus annuus L.*) belongs to the Compositae family of annual herbs, which are divided into edible the sunflower and oil sunflower (*Keeley, Cantley & Gallaher, 2021*). Sunflower oil is one of the world's four major oil crops (*Rele & Mohile, 2003*). The main uses of edible sunflower are ingestion and food addition (*Cao et al., 2018*). Because they have the advantages of simple cultivation and salt and drought tolerance, sunflowers are known as a pioneer crop in saline-alkali land (*Keeley, Cantley & Gallaher, 2021*; *Rele & Mohile, 2003*). Planting sunflowers on saline-alkali land has economic benefits as well as ecological benefits, such as the desalination of the soil (*Keeley, Cantley & Gallaher, 2021*; *Rele & Mohile, 2003*). Acting as a second messenger, calcium ions play an important role in the growth and development of plant cells (*Klimecka & Muszyńska, 2007*). When a plant is stimulated by the external environment, the $Ca^{2+}$ concentration changes in the cell are regulated through transport systems such as $Ca^{2+}$ channels. Corresponding signals are generated, including changes in ion transport, hormones, and protective enzyme systems, which then regulate the pore size, root hair development, and absorption of water and nutrients in plant guard cells (*Shukla et al., 2014*). Both abiotic and biotic stresses can induce plant stress resistance by activating the plant calcium ion pathway (*Shukla et al., 2014*; *Ju et al., 2021*). The calcium ion pathway is very important for the transduction of calcium signals, so the study of the calcium ion pathway has far-reaching significance when exploring the mechanism of plant resistance regulation (*Liu & Zhu, 1998*; *Xu et al., 2006*; *Xue et al., 2019*).

Calcium ion channels are mainly divided into hypotonic and hypertonic calcium ion channels, OSCA belonging to the latter (*Murthy et al., 2018*). OSCA is a kind of hyperosmotic stress sensor (also called a sensor protein) in *Arabidopsis thaliana*, as well as a $Ca^{2+}$-selective cation channel (*Zhang et al., 2018*). Other common calcium channel proteins (such as depolarization-activated $Ca^{2+}$-permeable channels (DACCs), hyperpolarization-activated $Ca^{2+}$-permeable channels (HACCs), and inositol 1,4,5-trisphosphate receptor (IP3R)) have been systematically studied in plants, but there have been few studies on OSCA (*Swarbreck, Colaço & Davies, 2013*; *Gao et al., 2021*). Using protein-conserved domain analysis, it was found that OSCA gene family proteins contained three conserved functional domains: late exocytosis (pfam13967), cytosolic domain of 10TM putative phosphate transporter (pfam14703), and calcium-dependent channel (pfam02714) (*Swarbreck, Colaço & Davies, 2013*; *Gao et al., 2021*). In crops such as *Arabidopsis thaliana* and rice, the OSCA gene family was divided into four subfamilies, all of which contained the DUF221 domain (*Yuan et al., 2014*; *Li et al., 2015*). The DUF221 domain is essential, representative, and decisive for both the OSCA gene family and represents the seven transmembrane domain regions of calcium-dependent channels that function as calcium channels in osmotic sensing (*Kiyosue, Yamaguchi-Shinozaki &*

*Shinozaki, 1994*). Previous studies used forward genetics to screen *AtOSCA1* as an unknown hyperosmotic calcium ion permeability channel (*Yuan et al., 2014*). The results of this study indicated that *OSCA1* may be an osmotic sensor in *Arabidopsis thaliana* (*Yuan et al., 2014*; *Zhang et al., 2020*). In rice, osmotic-related abiotic stresses (ABA, NaCl, and PEG) differentially induced the expression of 10 OsOSCA genes, of which *OsOSCA−3.1* was the identified *OsERD4* (early drought response) gene (*Li et al., 2015*). A drought-tolerant gene (*TaOSCA1.4*) was cloned in wheat that belongs to the same gene family as *AtOSCA1.8* and *OsOSCA1.4* and is related to grain number per ear and yield (*Zhai et al., 2020*). Thirteen GmOSCA genes in soybean (*Glycine max (Linn.) Merr*) were associated with drought and alkali stress responses (*Yin et al., 2021*). A membrane-integral protein (AtOSCA1.2) was obtained in *Arabidopsis thaliana* (*Liu, Wang & Sun, 2018*). This gene can increase the concentration of intracellular calcium ions under hyperosmotic stress and increase the permeability of the cell membrane to $K^+$ and $Na^+$. It has also been shown that *AtOSCA1.2* is an inherently mechanosensitive pore-forming ion channel that may be necessary for osmotic sensing, and *AtOSCA1.1* and *AtOSCA3.1* (also known as the *ERD4* gene) are mechanosensitive channels (*Liu, Wang & Sun, 2018*; *Zhang et al., 2018*).

OSCA family genes have been studied in a variety of plants and found to play potential roles in plant stress resistance (*Li et al., 2015*; *Yin et al., 2021*; *Tong et al., 2021*; *Miao et al., 2022*). Following the publication of the genomes of the sunflower and other Compositae plants, the evolution of this gene family in Compositae has not been systematically studied. In this study, the number, structure, collinearity, and phylogeny of the OSCA gene family were comprehensively analyzed using genomic data from the sunflower and six Compositae species. RNA-sequencing (RNA-seq) and qRT−PCR were used to analyze the expression of the sunflower OSCA gene in nine different tissues, six different hormones, and under NaCl stress conditions. The coexpression network of OSCA under salt stress was constructed using WGCNA. These results will further broaden our understanding of the evolutionary relationship of the sunflower OSCA gene family and provide a basis for its functional study under salt stress.

## MATERIAL AND METHODS

### Plant material

The sunflower salt-tolerant inbred line 19S05 (bred by Bayannur Institute of Agricultural and Animal Sciences) was selected for this study. This material had a plant height of 200 cm, number of leaves was approximately 30, leaves were green, without branches, and the inclination of the disc was 4. The diameter of the disc was 20 cm, yellow flowers were ligulated, and the weight of 100 seeds was 16 g. The grains were narrow oval, 2.00 cm long and 0.95 cm wide, with brown white edges, no streaks, and strong salt tolerance. Selected and full 19S05 seeds were washed with water, surface sterilized with 3% $H_2O_2$ for 10 min, and then placed in plastic pots filled with sterilized soil. After culturing four true leaves, a 200 mM NaCl stress treatment was performed. Root samples were taken at 0 h of treatment and then at 1 h, 3 h, 6 h, 12 h, and 24 h after treatment. Samples were then snap-frozen in liquid nitrogen (three biological replicates for each treatment) and stored in a −80 °C refrigerator.

## Identification and cioinformatics analysis of OSCA genes

*Helianthus annuus, Lactuca sativa var. Angustata, Gossypium hirsutum, Glycine max (Linn.) Merr, Arabidopsis thaliana, Oryza sativa, Vitis vinifera,* and *Zea mays* genomic and proteomic data were downloaded from the Ensembl Plants database (http://plants.ensembl.org/index.html) (*Bolser et al., 2016*). *Arctium lappa, Cichorium endivia, and Cichorium intybus* genomic and proteomic data were downloaded from the NCBI database (https://www.ncbi.nlm.nih.gov/). *Chrysanthemum morifolium* genomic and proteomic data were downloaded from the *Chrysanthemum* genome database (http://www.amwayabrc.com/zh-cn/). *Carthamus tinctorius L.* genomic and proteomic data were downloaded from the NGINX database (http://118.24.202.236:11010/filedown/). Additionally, we used the OSCA gene domain Hidden Markov model (PF02714) and HMMER software (http://hmmer.org/) to identify protein sequences validated in the NCBI-CDD (https://www.ncbi.nlm.nih.gov/cdd/) database (*Mistry et al., 2021*). ExPASy software (http://cn.expasy.org/tools) was used on the physicochemical data to calculate the protein, and EuLoc software performed subcellular localization prediction (*Wilkins et al., 1999*; *Chang et al., 2013*).

## Phylogenetic and collinear analysis of the OSCA gene family

The OSCA protein sequences from *Helianthus annuus, Arctium lappa , Chrysanthemum morifolium, Cichorium endivia, Cichorium intybus, Lactuca sativa var. Angustata, Carthamus tinctorius, Gossypium hirsutum, Glycine max (Linn.) Merr, Arabidopsis thaliana, Oryza sativa, Vitis vinifera,* and *Zea mays* were used for multiple sequence alignment. A phylogenetic tree was then constructed using the neighbor-joining method (the bootstrap method value was set to 1,000 and the remaining parameters were set to default values) (*Hall, 2013*). The nwk file was obtained using MEGA11 software, and a phylogenetic tree of the OSCA gene was made. The related collinearity map was constructed using the circle gene view function in TBtools (*Chen et al., 2020*; *Wang et al., 2012*).

## Chromosomal location, gene structure, and motif analysis

The chromosomal location information for the members of the sunflower OSCA gene family was extracted using TBtools software, and the chromosomal location map was drawn using Mapchart software (*Chen et al., 2020*; *Voorrips, 2002*). A phylogenetic tree of *Helianthus annuus, Arctium lappa L., Chrysanthemum morifolium, Cichorium endivia, Cichorium intybus, Lactuca sativa var. Angustata,* and *Carthamus tinctorius L.* OSCA sequences was constructed using MEGA11 software (*Wang et al., 2012*). OSCA motif types were analyzed using Multiple Expectation maximizations for Motif Elicitation (MEME: http://meme-suite.org/) (*Bailey et al., 2009*). Based on the mRNA sequence of the OSCA gene, the intron-exon structure was analyzed. Graphs were made using TBtools software (*Chen et al., 2020*).

## Analysis of cis-acting elements of the sunflower OSCA gene

The upstream 2,000 bp sequence of the sunflower OSCA gene was extracted, and the possible cis-acting elements were predicted using the PlantCARE database

(http://bioinformatics.psb.ugent.be/webtools/plantcare/html/). After screening, the TBtools software was used to map (*Chen et al., 2020*).

## RNA-seq analysis

Transcriptome sequencing of different tissues of sunflower under different hormone and salt stress conditions was downloaded from NCBI (SRP092742 and PRJNA866668). Fastp software was used for filtering, removing adapters, and quality control of the raw data. Hisat2 software aligned the filtered data to the sunflower reference gene, and StringTie software was used for expression quantification (*Kim et al., 2019*; *Pertea et al., 2015*). Gene expression was assessed using the FPKM method, and the expression data were normalized to log10 (FPKM+1). Then, the expression heatmap was drawn with TBtools software (*Chen et al., 2018*).

## qRT −PCR

Primers were designed in specific regions of the sunflower OSCA gene sequence using Primer 6.0 software. The *HaActin* gene was chosen as the reference gene for qRT-PCR analysis (Table S1; *Li & Brownley, 2010*). Root tissue cDNA was used as the template to detect the expression of candidate genes by qRT −PCR, and each sample was repeated three times. The total reaction system was 20 μl, qRT −PCR program: 94 °C for 30 s; 94 °C for 5 s, 60 °C for 15 s, 72 °C for 10 s for 38 cycles; and 4 °C for the end. Each group of data was replicated three times, and the $2^{-\Delta\Delta Ct}$ method was used for relative quantitative analysis (*Tanino et al., 2017*).

## WGCNA

The DEseq2 software package was used to identify differential genes under salt stress for WGCNA analysis. We selected $\beta = 6$ to process the original relation matrix in order to get a non-proportional adjacency matrix. The adjacency matrix was further transformed into a topological overlap matrix (TOM). The minimum number of genes in a module was 30, the threshold for merging similar modules was 0.25 (cutHeight =0.25), and the network type was "signed" (type ="signed" or networkType ="signed"). Cytoscape−3.8.2 software was used for visualization (*Shannon et al., 2003*). We performed Gene Ontology (GO) and Kyoto Encyclopedia of Genes and Genomes (KEGG) analyses using the cluster Profiler software package in R language.

# RESULT

## Identification of the OSCA gene family

We systematically studied the changes in the copy number of the OSCA gene family during the sunflower's evolution. First, using HMMsearch the OSCA gene was comprehensively searched from seven Compositae species (*Helianthus annuus, Arctium lappa, Chrysanthemum morifolium, Cichorium endivia, Cichorium intybus, Lactuca sativa,* and *Carthamus tinctorius*). The search results were validated against the NCBI-CDD database. *Helianthus annuus, Arctium lappa, Chrysanthemum morifolium, Cichorium endivia, Cichorium intybus, Lactuca sativa,* and *Carthamus tinctorius* contained

15, 17, 20, 19, 19, 21, and 14 OSCA sequences, respectively. We named them *HaOSCA1.1~HaOSCA3.1* according to the 15 sunflower sequences and the phylogenetic tree of 12 other plants. The protein encoded by sunflower OSCA family genes contained 648-838 amino acid residues, the length of the open reading frame (ORF) was 1,947–2,517 bp, the theoretical isoelectric point was 7.61−9.94, and the relative molecular weight was 73.42–96.62 kDa. The protein subcellular localization showed that HaOSCA was localized to the endoplasmic reticulum, vacuole, chloroplast, golgi apparatus. and mitochondrion (Table 1).

Fifteen HaOSCA genes were distributed on 10 sunflower chromosomes (Chr03, Chr04, Chr05, Chr06, Chr07, Chr09, Chr12, Chr15, Chr16, and Chr17) (Fig. 1). Among these, Chr07 and Chr16 staining contained the most (three) OSCA genes. Except for Chr10, which contained two OSCA genes, the other chromosomes contained only one OSCA gene.

## Evolutionary analysis of the OSCA gene family

To better understand the evolutionary relationship of the sunflower OSCA family genes, we constructed a phylogenetic tree of the full-length OSCA protein sequences of *Helianthus annuus*, *Gossypium hirsutum*, *Glycine max (Linn.) Merr*, *Arabidopsis thaliana*, *Oryza sativa*, *Vitis vinifera*, and *Zea mays* (Fig. 2A). According to the results, the evolutionary tree was divided into four groups. Compared with *Gossypium hirsutum*, *Glycine max (Linn.) Merr*, *Arabidopsis thaliana*, *Oryza sativa*, *Vitis vinifera*, and *Zea mays*, there was no sunflower OSCA protein in Group 4 (Fig. 2), which may have been caused by the loss of genes in the sunflower during the evolution process. The number of OSCA proteins in Group 1 and Group 2 subgroups was consistent both in the sunflower and in *Gossypium hirsutum*, *Glycine max (Linn.) Merr*, *Arabidopsis thaliana*, *Oryza sativa*, *Vitis vinifera*, and *Zea mays* (Fig. 2B). The number of OSCA genes in sunflower, *Arabidopsis thaliana*, and *Glycine max (Linn.) Merr* was basically the same, and the sunflower genes of the same branch were more closely related to those in *Arabidopsis thaliana* and *Glycine max (Linn.) Merr*. Except for soybean and cotton, the other crops in Group 3 all contained an OSCA sequence, which also indicated that the Group 3 subgroup of the OSCA gene family was more conserved during the plant's evolution. Although Group 3 and Group 4 had relatively few members, they were retained in the evolution of plants, suggesting that they may play important roles in biological processes. However, the loss of members of sunflower Group 4 may have also been caused by functional redundancy of genes, which requires further study.

To further explore the evolutionary relationship of the sunflower OSCA gene family, we selected the sunflower OSCA gene as the core and constructed a colinear relationship between sunflower and *Gossypium hirsutum*, *Glycine max (Linn.) Merr*, *Arabidopsis thaliana*, *Oryza sativa*, *Vitis vinifera*, and *Zea mays* OSCA genes (Fig. 3). We found that the sunflower OSCA had 18 collinear gene pairs with *Arabidopsis thaliana*, 36 with *Gossypium hirsutum*, 12 with *Glycine max (Linn.) Merr*, eight with *Oryza sativa*, and 10 with *Vitis vinifera*. There were 11 collinear gene pairs with *Zea mays* (Fig. 3). The greatest collinearity between sunflower OSCA and cotton may have been related to cotton having a larger genome and the largest number of OSCA genes. In addition to cotton, the sunflower OSCA

Shan et al. (2023), *PeerJ*, DOI 10.7717/peerj.15089

**Table 1 Sunflower OSCA gene family member information.**

| Gene name | Gene ID | Open reading frame/bp | Protein length/aa | Relative molecular weight (r)/kDa | Theoretical isoelectric point (pI) | Subcellular localization |
|---|---|---|---|---|---|---|
| HaOSCA1.1 | HannXRQ_Chr17g0558291 | 2,097 | 698 | 79.86 | 8.07 | Vacuole, Endoplasmic reticulum |
| HaOSCA1.2 | HannXRQ_Chr16g0516941 | 2,322 | 773 | 88.43 | 9.13 | Endoplasmic reticulum |
| HaOSCA1.3 | HannXRQ_Chr05g0129321 | 2,274 | 757 | 86.85 | 9.73 | Endoplasmic reticulum |
| HaOSCA1.4 | HannXRQ_Chr16g0506581 | 2,319 | 772 | 88.63 | 8.96 | Endoplasmic reticulum |
| HaOSCA1.5 | HannXRQ_Chr06g0164031 | 2,310 | 769 | 87.77 | 9.36 | Vacuole, Endoplasmic reticulum |
| HaOSCA1.6 | HannXRQ_Chr04g0118011 | 2,313 | 770 | 87.71 | 9.69 | Vacuole, Endoplasmic reticulum |
| HaOSCA1.7 | HannXRQ_Chr07g0194331 | 2,340 | 779 | 89.00 | 9.94 | Vacuole, Endoplasmic reticulum |
| HaOSCA1.8 | HannXRQ_Chr03g0081691 | 2,517 | 838 | 96.62 | 9.09 | Chloroplast, Endoplasmic reticulum |
| HaOSCA2.1 | HannXRQ_Chr17g0563981 | 1,947 | 648 | 73.42 | 8.54 | Chloroplast, Endoplasmic reticulum |
| HaOSCA2.2 | HannXRQ_Chr16g0525101 | 1,947 | 648 | 73.60 | 9.03 | Chloroplast, Endoplasmic reticulum |
| HaOSCA2.3 | HannXRQ_Chr07g0205641 | 2,172 | 723 | 82.82 | 8.75 | Golgi apparatus, Vacuole, Endoplasmic reticulum |
| HaOSCA2.4 | HannXRQ_Chr07g0188761 | 2,223 | 740 | 83.76 | 7.61 | Chloroplast, Endoplasmic reticulum |
| HaOSCA2.5 | HannXRQ_Chr09g0264291 | 2,133 | 710 | 80.64 | 9.1 | Golgi apparatus, Vacuole, Endoplasmic reticulum |
| HaOSCA2.6 | HannXRQ_Chr12g0379201 | 2,094 | 697 | 79.44 | 9.93 | Mitochondrion, Vacuole, Endoplasmic reticulum |
| HaOSCA3.1 | HannXRQ_Chr15g0481591 | 2,145 | 714 | 80.99 | 9.29 | Vacuole, Endoplasmic reticulum |
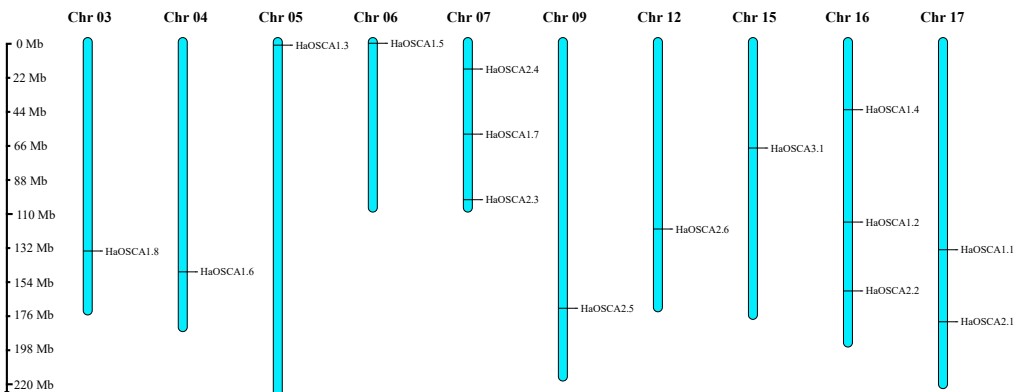

**Figure 1  Chromosomal location of the sunflower OSCA gene.**

had greater collinearity with *Arabidopsis thaliana* and *Glycine max (Linn.) Merr*, which was consistent with the results of the phylogenetic tree analysis.

To further study the evolutionary relationship of the sunflower OSCA gene family, we constructed a phylogenetic tree based on the OSCA protein sequences of six Compositae species (*Arctium lappa L., Chrysanthemum morifolium, Cichorium endivia, Cichorium intybus, Lactuca sativa var. Angustata,* and *Carthamus tinctorius L.*) and *Helianthus annuus* (Fig. 4A). Except for *Helianthus annuus* and *Carthamus tinctorius L.*, which lacked the Group 4 subgroup, all Asteraceae contained four subgroups (Fig. 4A). The *Cichorium endivia* and *Lactuca sativa var. Angustata* Group 4 subgroups contained three and four OSCA genes, respectively, while *Arctium lappa L., Chrysanthemum morifolium, Cichorium endivia*, and *Carthamus tinctorius L* all contained one, which was consistent with the results from *Arabidopsis thaliana, Glycine max (Linn.) Merr, Oryza sativa,* and *Zea mays* (Figs. 2B and 4B). Except for *Chrysanthemum morifolium* Group 3, which contained four sequences, there was no difference in the subgroups of Group 1, Group 2, and Group 3 based on Compositae.

We also constructed the collinear relationships for the OSCA of seven species of Compositae and found that there were three pairs of HaOSCA (*HaOSCA1.1* and *HaOSCA2.1, HaOSCA1.2* and *HaOSCA2.2, HaOSCA1.3* and *HaOSCA1.5*) with tandem repeats in sunflower genes (Fig. 5A), indicating that these three gene pairs may have been derived from tandem repeats. At the same time, five pairs of HaOSCA (*HaOSCA1.5* and *HaOSCA1.6, HaOSCA2.2* and *HaOSCA2.1, HaOSCA1.2* and *HaOSCA1.1, HaOSCA1.8* and *HaOSCA1.7, HaOSCA1.3* and *HaOSCA1.1*) showed very high identities (over 84.77%), implying that they may have arisen from segments or whole genes (Fig. 5A). Additionally, the ratio of non-synonymous to synonymous (Ka/Ks) between duplicate gene pairs was calculated, and the Ka/Ks values of all gene pairs were less than 1, indicating that these gene pairs may have undergone purifying selection (Table S2). *Helianthus annuus* had 32 collinear gene pairs with *Arctium lappa L.*, 32 with *Chrysanthemum morifolium*, 29 with *Cichorium endivia*, 31 with *Cichorium intybus*, and 39 with *Lactuca sativa var. Angustata,* and 18 collinear gene pairs with *Carthamus tinctorius L.*. In addition to having lower

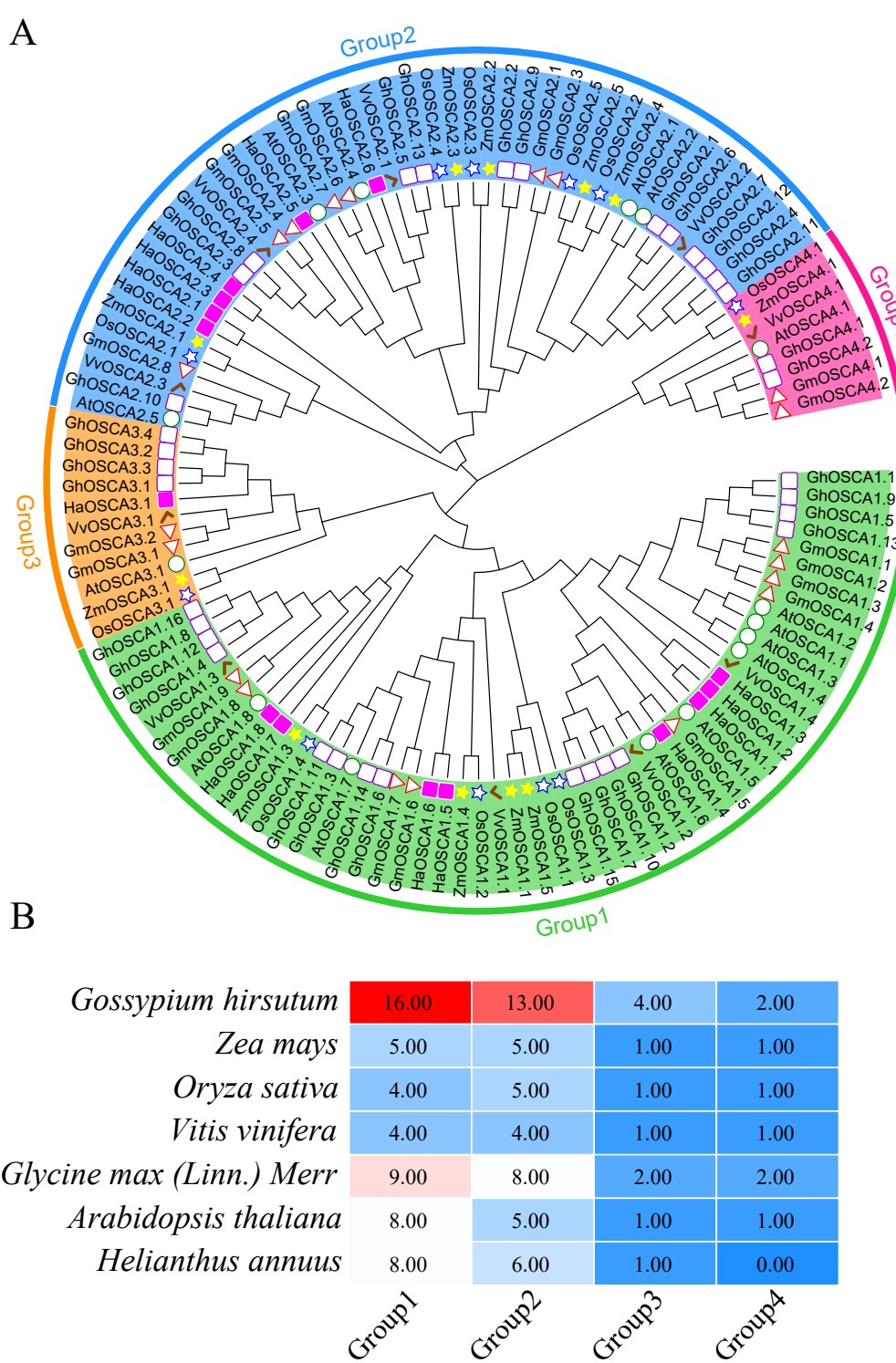

**Figure 2** (A) OSCA gene tree of sunflower, cotton, soybean, *Arabidopsis thaliana*, rice, grape, and maize. The different colors indicate different groups (1, 2, 3 and 4) of OSCA genes, different traits represent OSCA genes of different plants. (B) Sunflower, cotton, soybean, *Arabidopsis thaliana*, rice, grape, and maize OSCA gene evolutionary tree. The number of genes in each group.

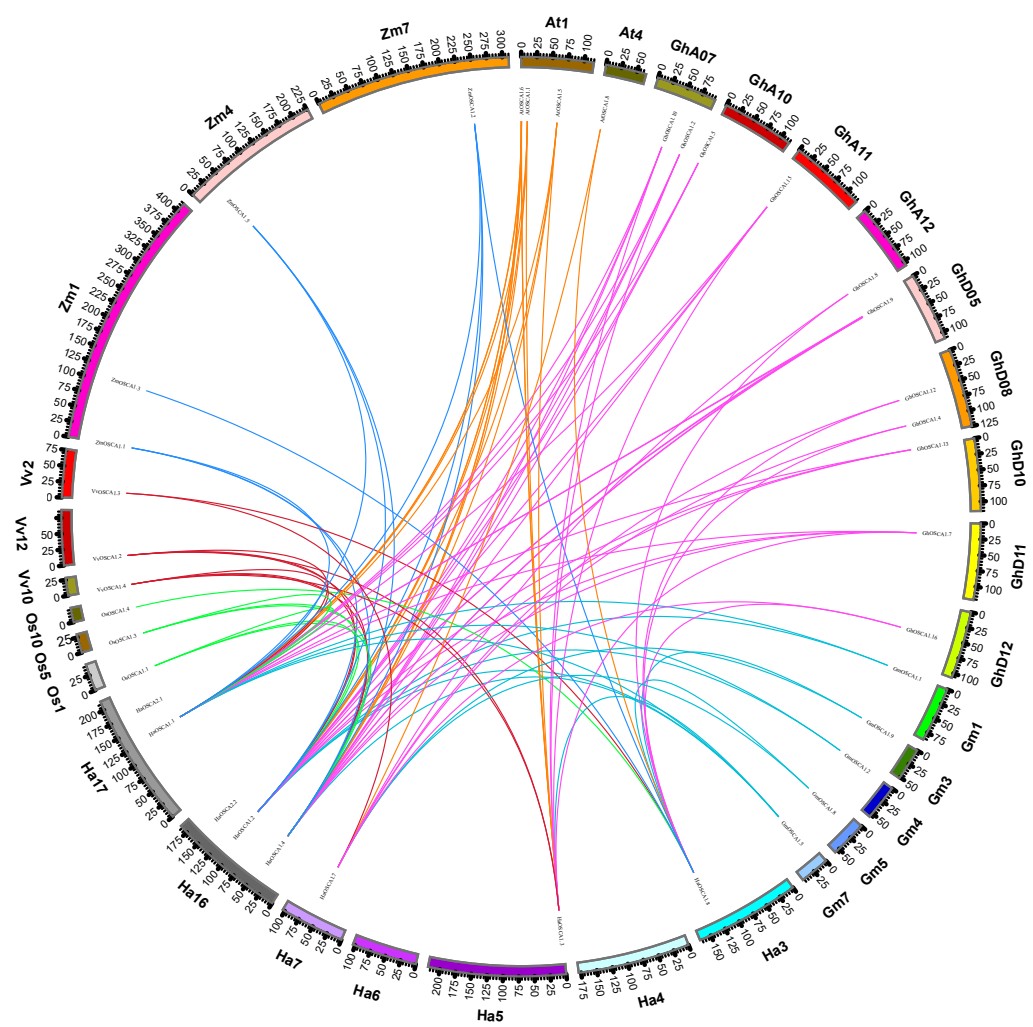

**Figure 3** **Collinear relationship of OSCA genes in sunflower, cotton, soybean, *Arabidopsis thaliana*, rice, grape, and maize.** The different colors represent a collinear relationship between different plants.

collinearity with *Carthamus tinctorius*, *Helianthus annuus* OSCA had more collinearity with other Compositae species, which indicates that the OSCA gene family was conserved in the evolution of Compositae. Moreover, *Helianthus annuus* and *Lactuca sativa var. Angustata* had the most collinear gene pairs and were also close to *Lactuca sativa var. Angustata* OSCA in the evolutionary tree, which indicates that the OSCA of *Helianthus annuus* and *Lactuca sativa var. Angustata* may have the same biological function.

## Gene structure and motif analysis of the OSCA gene of Compositae

To further understand the composition of the sunflower OSCA gene, we compared the OSCA gene structure of seven Compositae species. All OSCA genes, except Group 4, contained introns (Fig. 6). Members of the same subfamily had similar conserved motifs, and Group 4 had the simplest gene structure and least number of motifs (Fig. 6). The Pfam website (http://pfam.xfam.org/family/pf02714#tabview=tab0) showed that the RSN1_7TM

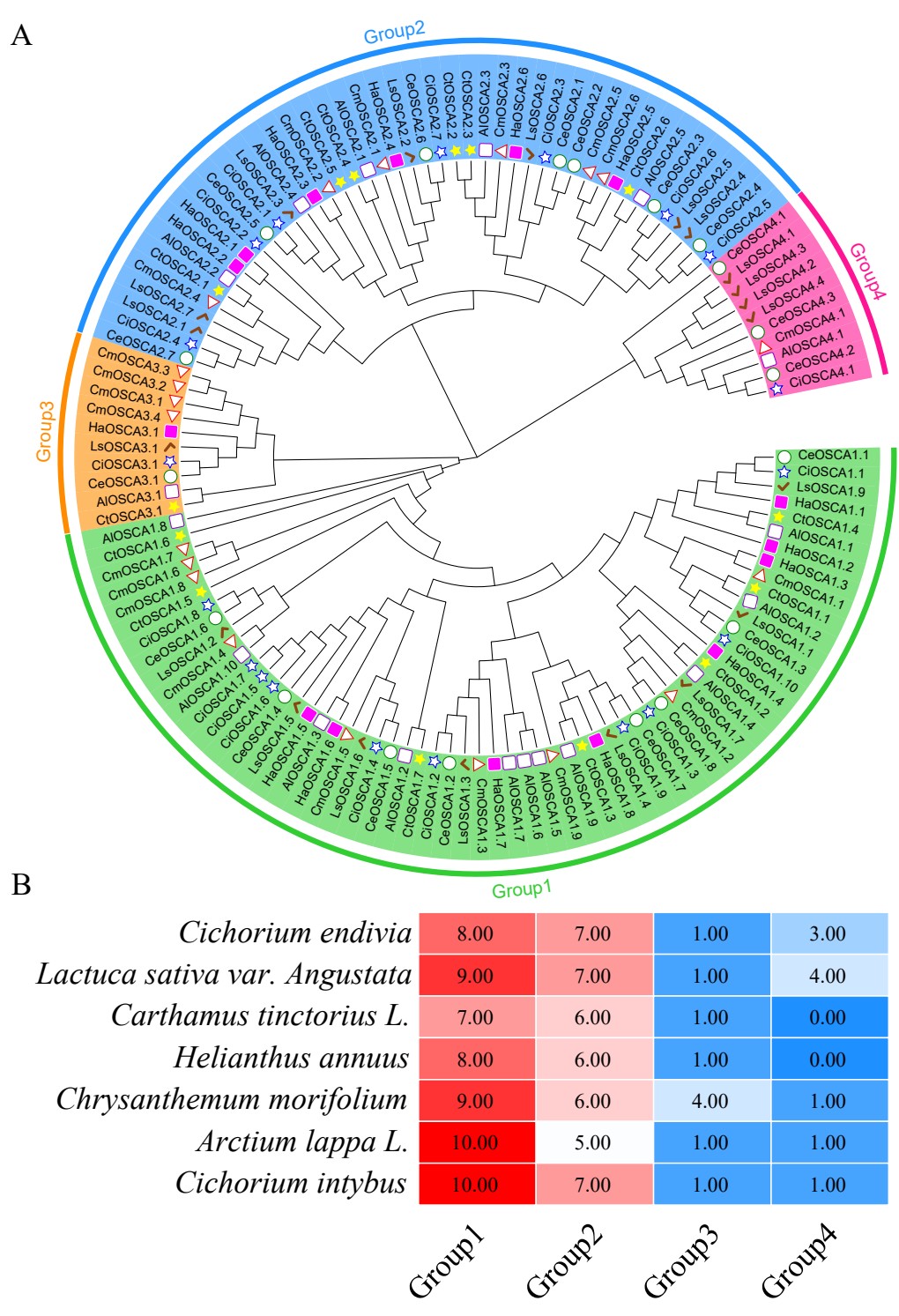

**Figure 4** (A) OSCA gene tree of sunflower, *Arctium lappa, Chrysanthemum morifolium, Cichorium endivia, Cichorium intybus, Lactuca sativa,* and *Carthamus tinctorius*. The different colors indicate different groups (1, 2, 3 and 4) of OSCA genes, different traits represent OSCA genes of different plants. (B) Sunflowers, *Arctium lappa, Chrysanthemum morifolium, Cichorium endivia,* and Cichori.

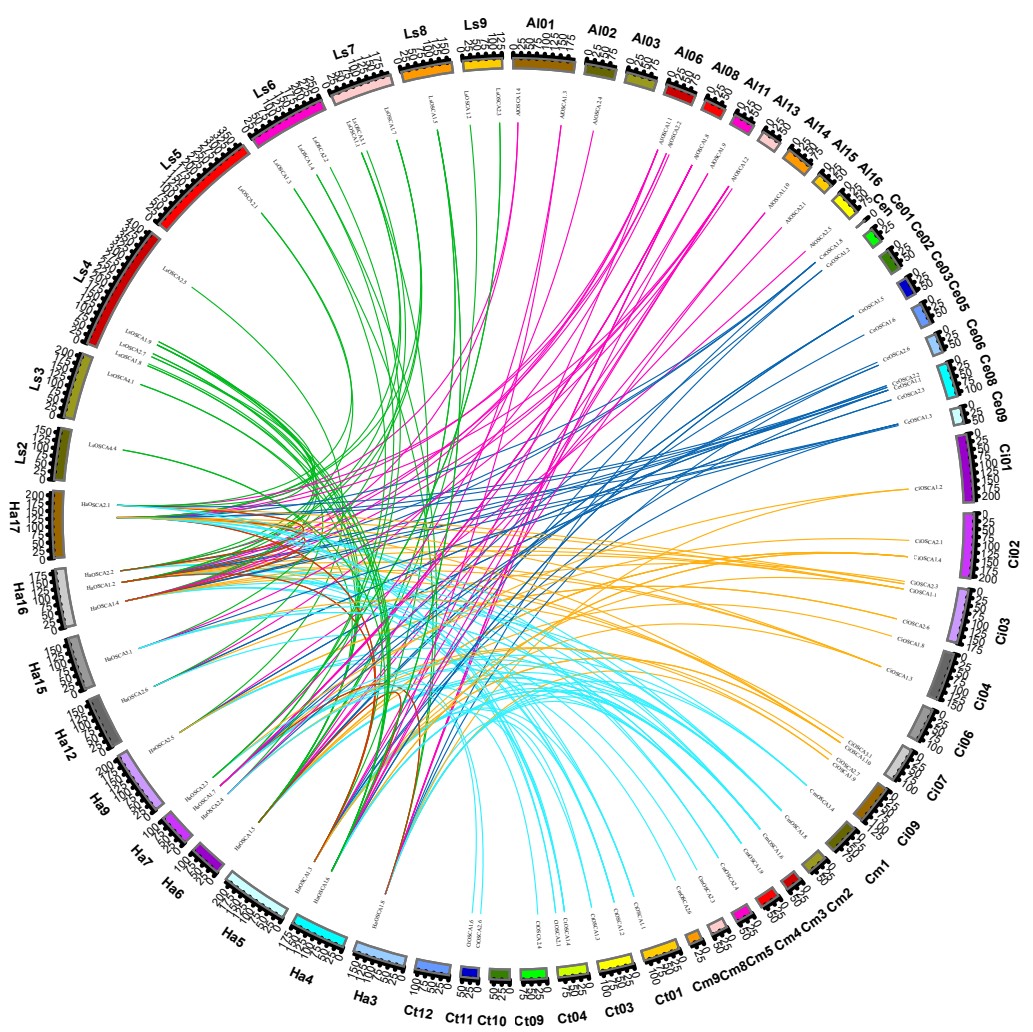

**Figure 5** **Collinearity of sunflowers, *Arctium lappa, Chrysanthemum morifolium, Cichorium endivia, Cichorium intybus, Lactuca sativa,* and *Carthamus tinctorius* OSCA genes.** The different colors represent a collinear relationship between different plants.

domain generally contained seven transmembrane domains. In this article, motif 2, motif 3, motif 4, motif 6, motif 7, motif 8, and motif 9 showed that they may contain multiples of the seven transmembrane domains. Only a few OSCA genes (*CtOSCA1.5, CiOSCA2.2,* and *LsOSCA4.1*) contained a motif, although they were relatively long. *CmOSCA2.2* was the longest and contained 10 exons, but it also contained eight motifs. The *LsOSCA4.4* gene had the shortest length and contained one exon but only one motif (Motif 3). This indicates that the distribution of OSCA gene motifs in Compositae was highly correlated with the length of the genes.

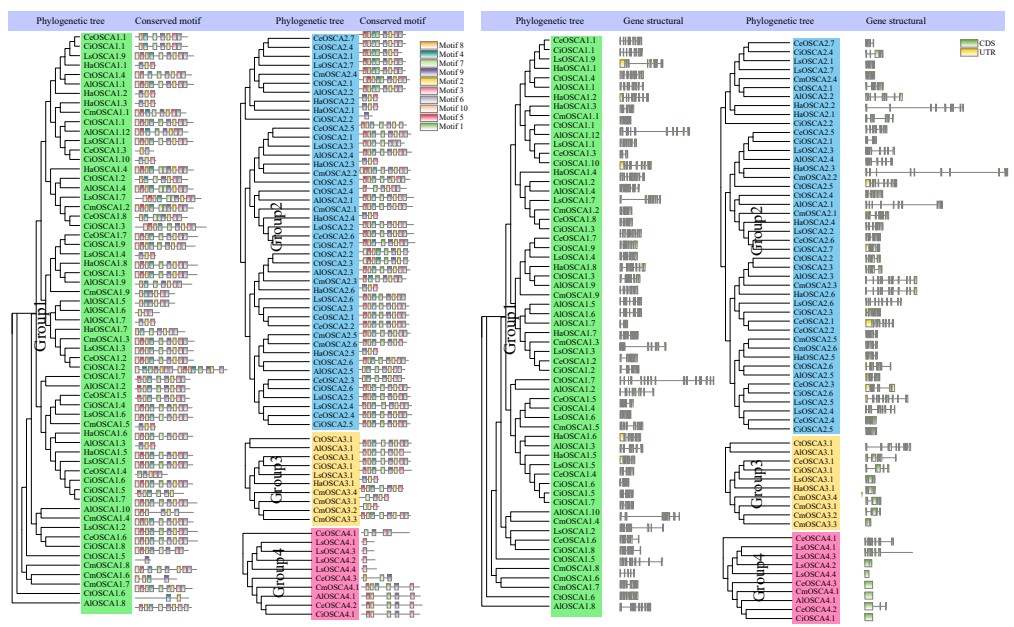

**Figure 6 Phylogenetic tree, gene structure, and conserved motif analysis of the OSCA gene family of Compositae.** Different motifs are displayed in different colored boxes as indicated on the right side.

## Analysis of cis-acting elements in the promoter of the sunflower OSCA gene

Transcription factors (TFs) are proteins that can bind to the promoter regions of genes in a sequence-specific manner in order to regulate transcription, thereby regulating plant functions, including responses to environmental factors, growth, and development (*Warren, 2002*). Analysis of the 2,000 bp upstream sequence of the sunflower OSCA gene start codon showed that there were multiple stress and hormone response elements in the upstream sequence of the HaOSCA gene, with different types and numbers (Fig. 7). Three main categories of cis elements were found in the promoter sequences of HaOSCA genes. The first category was involved in phytohormones, such as abscisic acid (ABA), jasmonic acid (JA), auxin, gibberellins (GA), and salicylic acid (SA). The second category was associated with stresses, such as low-temperature responsiveness, defense, and stress responsiveness. The last category was mainly MYB binding sites. Importantly, all 14 HaOSCA genes, except *HaOSCA1.8*, contained the JA-responsive element (TGACG-motif and CGTCA-motif), and the ABA responsive element (ABRE) was found in almost all gene promoters (Table S3). Interestingly, cis-acting elements were not found in the promoter region of *HaOSCA1.8*. These results showed that HaOSCA may affect hormone signal responsiveness and stress adaptation. No cytokinin-responsive elements were identified in the HaOSCA promoter regions.

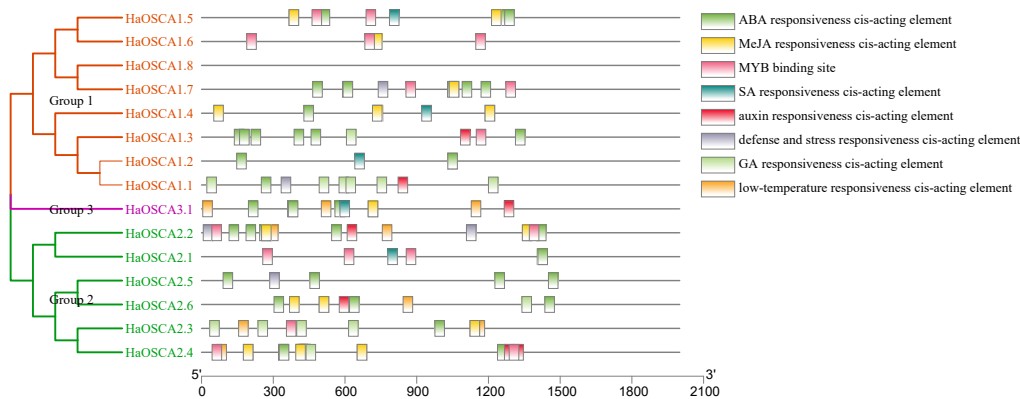

**Figure 7** **Analysis of homeopathic elements in the promoters of the sunflower OSCA gene family.**

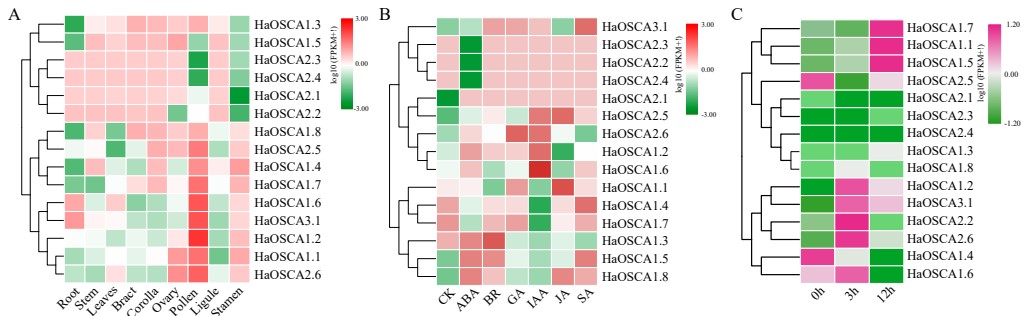

**Figure 8** **Expression analysis of the sunflower OSCA gene.** (A) Heatmap of expression patterns in different tissues. (B) Heatmap of expression patterns under different hormone treatments. (C) Heatmap of expression patterns under salt stress.

## Expression analysis of the sunflower OSCA gene

Gene expression patterns are usually closely related to gene functions. To gain insight into the expression patterns of sunflower OSCA genes, we first analyzed the expression patterns of 15 sunflower OSCA genes in nine tissues (Fig. 8A). The results showed that there were nine sunflower OSCAs (*HaOSCA1.1, HaOSCA1.2, HaOSCA1.4, HaOSCA1.6, HaOSCA1.7, HaOSCA1.8, HaOSCA2.5, HaOSCA2.6,* and *HaOSCA3.1*) in pollen and stamen. The expression was higher than in other tissues, and the remaining six had higher expression levels in root, stem, leaves, bract, corolla, ovary, and ligament, which indicated that sunflower OSCA had a wide range of tissue expression types.

Plant hormones play an important role in the processes of growth, development, and stress. We analyzed the expression pattern of the sunflower OSCA gene induced by six hormones (ABA, BR, JA, GA, IAA, and SA) (Fig. 8B). Six genes (*HaOSCA1.3, HaOSCA1.5, HaOSCA1.8, HaOSCA2.2, HaOSCA2.3,* and *HaOSCA2.4*) were more strongly induced by ABA than other hormones, six genes (*HaOSCA1.2, HaOSCA1.4, HaOSCA1.6, HaOSCA1.7,*

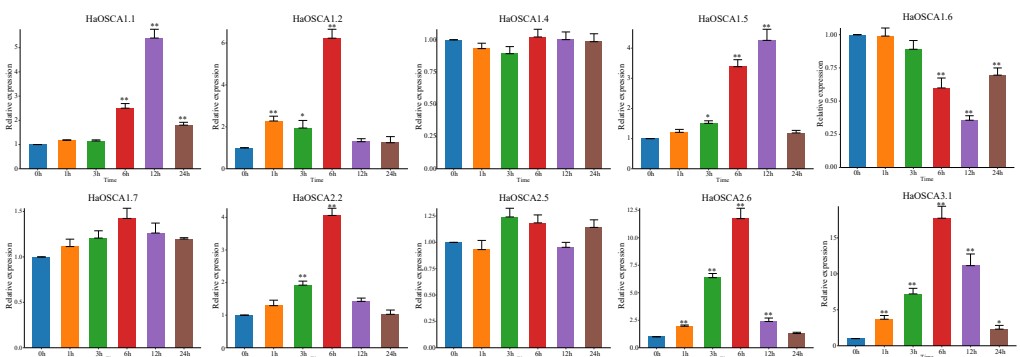

**Figure 9 Expression analysis of the sunflower OSCA gene under NaCl stress.** Error bars represent the average of three replicates ± Sd. (* $P < 0.05$; ** $P < 0.01$). The housekeeping gene *HaActin* was used as the internal reference gene.

*HaOSCA2.5,* and *HaOSCA2.6*) were more strongly affected by IAA than other hormones, and HaOSCA1.1 was most strongly affected by JA.

Next, we analyzed the expression pattern of the sunflower OSCA gene under salt stress. All sunflower OSCA gene expression levels changed after NaCl treatment (Fig. 8C). Seven OSCA genes (*HaOSCA1.1, HaOSCA1.2, HaOSCA1.5, HaOSCA1.7, HaOSCA2.2, HaOSCA2.6,* and *HaOSCA3.1*) were upregulated, three (*HaOSCA1.4, HaOSCA1.6,* and *HaOSCA2.5*) were significantly downregulated, and the expression levels of the remaining five (*HaOSCA1.3, HaOSCA1.8, HaOSCA2.1, HaOSCA2.3,* and *HaOSCA2.4*) also changed, but not significantly (Fig. 8C). These results indicated that the 10 OSCA genes (*HaOSCA1.1, HaOSCA1.2, HaOSCA1.4, HaOSCA1.5, HaOSCA1.6, HaOSCA1.7, HaOSCA2.2, HaOSCA2.5, HaOSCA2.6,* and *HaOSCA3.1*) may play a role in the sunflower's response to salt stress.

## qRT−PCR of the sunflower OSCA gene under salt stress

With the increase of soil salinization, salt stress has become the most important abiotic stress faced by the sunflowers. Based on the previous expression analysis, we found that 10 genes (*HaOSCA1.1, HaOSCA1.2, HaOSCA1.4, HaOSCA1.5, HaOSCA1.6, HaOSCA1.7, HaOSCA2.2, HaOSCA2.5, HaOSCA2.6,* and *HaOSCA3. 1*) may be involved in the response of sunflower to salt stress. qRT −PCR was used to study the expression patterns of these 10 genes under NaCl stress. Compared with levels before stress (0 h), the expression of seven genes (*HaOSCA1.1, HaOSCA1.2, HaOSCA1.5, HaOSCA1.6, HaOSCA2.2, HaOSCA2.6,* and *HaOSCA3.1*) was significantly changed at different points in time during treatment (Fig. 9). *HaOSCA1.6* was significantly downregulated and the remaining six genes were significantly upregulated. The expression levels of *HaOSCA2.6* and *HaOSCA3.1* reached more than a 10-fold change at 6 h of NaCl stress. These results suggest that *HaOSCA2.6* and *HaOSCA3.1* may play an important role in the response of sunflower to salt stress.

## Protein interaction of sunflower OSCA gene

Genes typically perform their biological functions and signal transduction pathways through interacting networks, so studying the underlying interaction networks associated with gene families can provide a better understanding of their functions. To elucidate the role of OSCA genes in salt stress in the sunflower, we used WGCNA to analyze OSCA genes (*HaOSCA1.1, HaOSCA1.2, HaOSCA1.5, HaOSCA1.6, HaOSCA2.2, HaOSCA2.6,* and *HaOSCA3.1*) and the salt coexpression network of all differentially expressed genes under stress. Genes were clustered using dynamic shearing and divided into modules. By calculating the eigenvectors of each module and merging similar modules, a total of seven gene coexpression modules were obtained (Fig. 10A). In this study, *OSCA* genes in each module were selected as core genes, and gene interaction network diagrams were drawn using core genes and their interacting genes (Fig. 10B and Table S4). To further uncover the function of the network, we performed an enrichment analysis of all interacting genes (Figs. 10C and 10D). The GO enrichment results showed that the biological processes involved were mainly cellular component organization or biogenesis, biological regulation, multicellular organismal process, response to stimulus and cellular process, cellular components are extracellular regions, cells and extracellular regions, molecular functions are transporter activities, antioxidant activities, transcription regulator activities, and molecular carrier activity (Fig. 10C). The KEGG pathway was mainly alpha-linolenic acid metabolism, glycerolipid metabolism, peroxisome, and tryptophan metabolism (Fig. 10D). This network further demonstrated the complex functions and potential roles of the OSCA gene family in sunflower salt-stressed species. The results of this study lay a foundation for further research on the functions and molecular mechanisms of these genes in the sunflower under salt stress and provide a reference for further study of the sunflower OSCA gene family.

## DISCUSSION

Over their long evolutionary history, all living plants are thought to have undergone at least one or more whole genome duplication (WGD) events, which provide opportunities for repeated genes to acquire functional diversification, leading to more complex organisms. In this process of differentiation, the gene family produces more members (*Zhang et al., 2018*; *Swarbreck, Colaço & Davies, 2013*). Plant biologists have always been fascinated by the structure, function, and evolution of genes, and the interactions and adaptations between the environment and plants have been well studied (*Gao et al., 2021*; *Yuan et al., 2014*). With the continuous publication of plant genomes, the structure, function, and evolution of various gene families have been extensively studied in plants, and these results can provide more insight into the origin, diversity, and biological function of these gene families (*Kiyosue, Yamaguchi-Shinozaki & Shinozaki, 1994*; *Glasauer & Neuhauss, 2014*; *Li et al., 2015*; *Tong et al., 2021*; *Miao et al., 2022*). Asteraceae is made up of many different species, each with a rich morphological type that exhibits extreme characteristics (*Badouin et al., 2017*). The evolution of the cultivated sunflower progressed in two phases (domestication by native North Americans, followed by breeding according to traits related

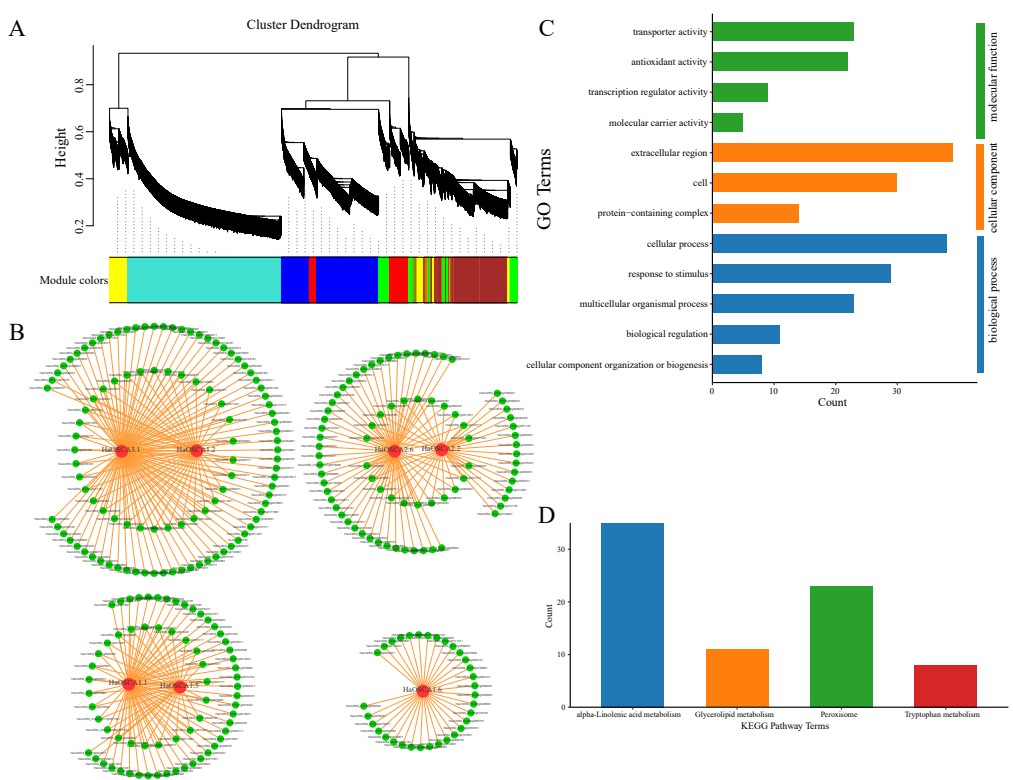

**Figure 10** (A) Module partition of WGCNA clustering. (B) Sunflower OSCA coexpression network. (C) GO enrichment analysis results of coexpression genes. (D) KEGG pathway enrichment analysis results of coexpression genes.

to modern agricultural production) and have two complex WGD histories that provide an excellent opportunity to study the relationship between gene family isolation and plant morphological changes (*Badouin et al., 2017*). In a breakthrough study in 2014, the first hypertonic-induced $Ca^{2+}$ conductive cation channel obtained from plants was discovered, which was designated to reduce hypertonic-induced $[Ca^{2+}]$ increase1 (*OSCA1*) (*Yuan et al., 2014*). *OSCA1* mutants exhibited impaired osmotic $Ca^{2+}$ signaling in guard cells and root cells, as well as weakened water transport regulation and root growth response under osmotic stress conditions (*Yuan et al., 2014*). This work advanced the study of OSCA genes, leading to the identification of more OSCA gene families (*Yuan et al., 2014*). The 15 sunflower OSCA genes obtained in this study were close to the number of OSCA gene families in other crops, but much fewer than in soybeans and cotton, which may be related to the size of the soybean and cotton genomes and the large number of replications within their OSCA family members. The cell membrane (the dividing line between the internal and external environment of the cell) is the first to feel the stress stimulus, and OSCA is a calcium channel with multiple transmembrane domains, which is consistent with the subcellular structure predictions in this study (Table 1). This may be because when plants encounter adversity, genes in cells will regulate multiple organs to, through the synthesis
of organic matter, generate energy and jointly resist adversity and stress, which needs to be further explored (*Yin et al., 2021*; *Miao et al., 2022*).

In *Arabidopsis thaliana*, two tandem repeat events occurred in the OSCA family (*Kiyosue, Yamaguchi-Shinozaki & Shinozaki, 1994*). Moreover, tandem duplications of the OSCA gene family have been detected in rice, litchi, wheat, and soybean species (*Li et al., 2015*; *Tong et al., 2021*; *Yin et al., 2021*; *Miao et al., 2022*). In this study, an in-depth evolutionary analysis of the sunflower OSCA gene family was performed using the OSCA genes of 13 species (Figs. 2–6). The HaOSCA evolutionary tree was subdivided into three subfamilies, which was inconsistent with the OSCA family member evolutionary tree subfamily division of rice, soybean, corn, and other plants (*Li et al., 2015*; *Tong et al., 2021*; *Ding et al., 2019*; *Wang et al., 2019*). HaOSCA has the closest evolutionary tree clades and the largest number of collinear gene pairs to *Arabidopsis thaliana* and soybean, suggesting that the research basis of *Arabidopsis thaliana* and soybean OSCA genes can be used to expand our understanding of sunflower OSCA genes (*Li et al., 2015*; *Tong et al., 2021*; *Ding et al., 2019*; *Wang et al., 2019*). Three pairs of tandem duplication genes were found in the sunflower OSCA gene family, which indicated that the expansion and evolution of the sunflower OSCA gene family might be caused by tandem duplication. According to the Ka/Ks analysis, the strong purifying selection signal also indicated the functional importance of the sunflower OSCA gene. We also hypothesized that this may be because nonsynonymous substitutions generally bring harmful traits and rarely result in an evolutionary advantage (*Swarbreck, Colaço & Davies, 2013*; *Miao et al., 2022*). The six Asteraceae species contained four subgroups, and only three of these subgroups (1, 2, and 3) were included in the sunflower. Previous studies in tomatoes found that tomato OSCA family members were divided into five subfamilies, which indicates that the number of genetic family evolutionary subfamilies may be greatly related to crop growth habits (*Wang et al., 2019*). We found that sunflower OSCA had more collinearity with other Asteraceae plants and less collinearity with Carthamus tinctorius, indicating that the OSCA gene family was conserved during the evolution of Asteraceae. Sunflower and *Lactuca sativa var. Angustata* had the most colinear gene pairs, and the evolutionary tree was also closer to *Lactuca sativa var. Angustata* OSCA, suggesting that sunflower may have the same biological function as *Lactuca sativa var. Angustata* OSCA. Combined with the evolutionary tree, the number of CDS coding regions of family members from the same subfamily was similar, but the number of different subfamilies varied greatly, and it is speculated that the functions performed by subfamilies may be different.

The functions of multiple OSCA genes involved in regulating plant growth, development, and various stress responses have been verified, and the roles played by different tissues and growth stages are different (*Zhang et al., 2018*; *Swarbreck, Colaço & Davies, 2013*; *Gao et al., 2021*; *Kiyosue, Yamaguchi-Shinozaki & Shinozaki, 1994*; *Yin et al., 2021*; *Miao et al., 2022*). In this study, cis-acting elements in the promoter region of the sunflower OSCA gene were analyzed, and we found that cis-acting elements in the binding sites of folic ABA and JA-bound cis-acting elements were the most abundant. Through expression analysis, we also found that more HaOSCA genes were induced by ABA and JA (Fig. 8B). A growing number of studies have found that ABA is the most important hormone for plant abiotic

stress, and JA also plays an important role in this process (*Wang et al., 2020*; *Baldoni, Genga & Cominelli, 2015*). ABA has been found to activate the phosphorylation membrane of SnRKs bound to NADPH oxidase RBOHF, increasing the production of $H_2O_2$ in apoplasts (*Umezawa et al., 2009*). Salt-induced ABA and $Ca^{2+}$ signaling activates RBOHF activity through *SnRK2.6* and *CIPK11/26* signaling modules, while *ABI1* is inhibitory. ABA, ROS, and $Ca^{2+}$ exhibit complex signal crosstalk to control plant responses to salt stress (*Quan et al., 2007*). JA levels are elevated and JA signaling is activated under salt stress (*Chen et al., 2016*). The F-box protein COI1, also known as the JA receptor, forms complexes with SKP1 and CULLIN1 to mediate JAZ degradation (*Valenzuela et al., 2016*). When JAZ is removed, inhibited transcription factors such as MYC activate the expression of JA response genes. Activation of JA signaling under salt stress eventually leads to inhibition of primary root growth. Exogenous JA can mitigate salt toxicity by maintaining ROS or ionic homeostasis (*Valenzuela et al., 2016*). These combined results hint at the complexity and importance of the sunflower OSCA gene in the coordination of growth and adversity.

The yield and quality of the sunflower are susceptible to various biotic and abiotic stresses (*Dar et al., 2021*; *Gogna & Bhatla, 2019*). Abiotic stresses such as salinity and temperature often have a greater impact on plants (*Gong et al., 2020*; *Verma, Ravindran & Kumar, 2016*). Although sunflowers are not halophytes, they have a strong ability to tolerate salt and alkali (*Keeley, Cantley & Gallaher, 2021*; *Rele & Mohile, 2003*). This candidate gene for mining salt tolerance-related genes is a good research model. The soybean OSCA gene family generally responds to abiotic stress (saline-alkali), and nine OSCA genes were also upregulated or downregulated in rice abiotic stress treatments (*Li et al., 2017*). In the sunflower, we found that the expression levels of *HaOSCA1.1, HaOSCA1.2, HaOSCA1.5, HaOSCA1.6, HaOSCA2.2, HaOSCA2.6,* and *HaOSCA3.1* all changed significantly at different time points during NaCl treatment. Of these, only *HaOSCA1.6* was significantly downregulated (Fig. 9). The expression levels of *HaOSCA2.6* and *HaOSCA3.1* reached more than a 10-fold change at 6 h of NaCl stress. These results suggest that *HaOSCA2.6* and *HaOSCA3.1* may be the most important in the sunflower's response to salt stress.

Studies on other plants have shown that most processes related to plant growth, development, and physiology are regulated by networks of protein−protein interactions (*Yuan et al., 2014*; *Zhang et al., 2020*; *Liu, Wang & Sun, 2018*). Building interaction networks between proteins is important for understanding how proteins work in biological systems. It is of great significance to understand the growth and development of plants, biological signals, and response mechanism of energy metabolism in some specific physiological states, as well as to understand the functional connections between proteins. We used WGCNA to analyze the coexpression network of sunflower OSCA genes and differentially expressed genes under salt stress (Fig. 10). Using the KEGG pathway enrichment analysis of the interacting genes, we found that the main enrichments were in the alpha-linolenic acid metabolism, glycerolipid metabolism, and peroxisome and tryptophan metabolism pathways (Fig. 10D). In *Arabidopsis thaliana*, alpha-linolenic acid metabolism has been found to be important in synthesizing JA (*Valenzuela et al., 2016*). *AtLOX3* is involved in the process of *Arabidopsis thaliana* salt stress (*Kilaru et al., 2011*). Constitutive overexpression of AOC in wheat led to elevated JA levels and also confirmed JA

involvement in salt stress response (*Kilaru et al., 2011*). The function of JAZ congeners in salt stress response has been demonstrated in many plant species (*Valenzuela et al., 2016*). In addition, the bHLH factor RICE SALT SENSITIVE3 (*RSS3*)/*OsbHLH094* has been found to interact with JAZ to regulate salt tolerance (*Toda et al., 2013*). In addition, overexpression of *OsbHLH148* in rice mutants induced high expression of OsDREB and OsJAZ under drought stress (*Seo et al., 2011*). Similarly, overexpression of *OsJAZ9* significantly improved the salinity and drought tolerance of rice (*Seo et al., 2011*). However, whether OSCA relies on the JA pathway to improve the salinity tolerance of plants has not been reported, but provides a reference for our future research. In conclusion, the expression analysis of sunflower OSCA genes under salt stress and the construction of a coexpression network will provide clues to further study the salt tolerance of sunflower.

## CONCLUSION

In this study, a systematic evolutionary analysis of sunflower OSCA genes was performed using 13 plant OSCA genes, and the sunflower OSCA gene family was found to be conserved in the evolution of Compositae. The analysis of the expression patterns of six hormones and salt stress in nine tissues showed that the OSCA gene played an important role in regulating the response of sunflower to salt stress, and *HaOSCA2.6* and *HaOSCA3.1* were the most important in the sunflower's response to salt stress. The coexpression network of the sunflower OSCA gene under salt stress was also constructed based on WGCNA. In this study, the systematic evolution and expression analysis of the sunflower OSCA gene family was carried out for the first time, and provided clues for further research on the role of sunflower OSCA genes in salt-tolerant species.

### Funding
This research was supported by grants from the Plant Variety Protection Foundation of the Ministry of Agriculture and Rural Affairs (2130135DUS202104). The funders had no role in study design, data collection and analysis, decision to publish, or preparation of the manuscript.

### Grant Disclosures
The following grant information was disclosed by the authors:
Plant Variety Protection Foundation of the Ministry of Agriculture and Rural Affairs: 2130135DUS202104.

### Competing Interests
The authors declare there are no competing interests.

### Author Contributions
- Feibiao Shan conceived and designed the experiments, performed the experiments, analyzed the data, prepared figures and/or tables, authored or reviewed drafts of the article, and approved the final draft.

- Yue Wu performed the experiments, analyzed the data, prepared figures and/or tables, and approved the final draft.
- Ruixia Du performed the experiments, analyzed the data, prepared figures and/or tables, and approved the final draft.
- Qinfang Yang performed the experiments, authored or reviewed drafts of the article, and approved the final draft.
- Chunhui Liu performed the experiments, prepared figures and/or tables, and approved the final draft.
- Yongxing Wang conceived and designed the experiments, performed the experiments, authored or reviewed drafts of the article, and approved the final draft.
- Chun Zhang conceived and designed the experiments, authored or reviewed drafts of the article, and approved the final draft.
- Yang Chen conceived and designed the experiments, performed the experiments, analyzed the data, prepared figures and/or tables, and approved the final draft.

## Data Availability

The qRT-PCR raw data is available in the Supplemental File.

## Supplemental Information

Supplemental information for this article can be found online at http://dx.doi.org/10.7717/peerj.15089#supplemental-information.

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
