# Peer review of "Evolutionary analysis of the OSCA gene family in sunflower (Helianthus annuus L) and expression analysis under NaCl stress"

_PeerJ, doi:10.7717/peerj.15089_

## Round 0.1 · original submission · Major Revisions

Dear Author,
Please see the comments by Reviewers 2 and 3. Though Reviewer 1 has suggested rejecting the manuscript, I recommend its acceptance after Major revision. Please follow the deadline for the revised manuscript.
Thanks

Reviewer 1 ·

Basic reporting

no comment

Experimental design

no comment

Validity of the findings

no comment

Additional comments

The authors present a bioinformatics and expression analysis of the OSCA gene family in sunflower (Helianthus annuus L).
1. In Introduction section, the authors don’t describe the structure and function of OSCAs in detail. It seems that the authors have little knowledge of OSCAs. In addition, many relevant papers in other plants were even not mentioned, such as OSCAs in maize (Cao et al. 2020), and wheat (Tong et al. 2022), and cucumber (Yang et al. 2022).
2. PF03514 is the hidden markov model (HMM) profile of GRAS TF. Therefore, I was doubted the results of identified Sunflower OSCA family genes in this study. By the way, no group 4 OSCA was identified in Sunflower.
3. Line 112: multiple sequence alignment was performed by which tool?
4. Line 120: the description of evolutionary tree was repeated to line 114-117.
5. Since OSCAs are transmembrane proteins usually consist of 11 transmembrane helices and a cytosolic soluble domain, however, all Sunflower OSCAs were localized in the Endoplasmic reticulum by subcellular localization prediction?
6. The structure of results is vague and out-of-order, the authors should analyze their data in more useful ways and present the most useful data.
7. The qRT-PCR data of several HaOSCA genes were not consistent with the results of RNA-seq. How can the authors explain it?
8. There were huge grammars mistakes present in this manuscript, such as open reading frame (OFR).

References
Cao L, Zhang P, Lu X, Wang G, Wang Z, Zhang Q, Zhang X, Wei X, Mei F, Wei L, Wang T. Systematic analysis of the maize OSCA Genes revealing ZmOSCA family members involved in osmotic stress and ZmOSCA2.4 confers enhanced drought tolerance in transgenic Arabidopsis. Int J Mol Sci 2020; 21 (1): 351.
Tong K, Wu X, He L, Qiu S, Liu S, Cai L, Rao S, Chen J. Genome-wide identification and expression profile of OSCA gene family members in Triticum aestivum L. Int J Mol Sci 2022; 23 (1): 469.
Yang S, Zhu C, Chen J, Zhao J, Hu Z, Liu S, Zhou Y. Identification and expression profile analysis of the OSCA gene family related to abiotic and biotic stress response in cucumber. Biology (Basel) 2022; 11 (8): 1134.

Reviewer 2 ·

Basic reporting

This is a good study focusing new part of sunflower genome. Basic mistakes have been seen and pointed for correction. Please see pdf for details

Experimental design

Its ok.

Validity of the findings

Findings are valid.

Additional comments

See Pdf For detailed comments

Annotated reviews are not available for download in order to protect the identity of reviewers who chose to remain anonymous.

Reviewer 3 ·

Basic reporting

This manuscript reports the identification and analysis of OSCA gene family in sunflower. The manuscript is structured in an acceptable research paper format and includes sufficient background and appropriate references. The figures and tables in this manuscript are of high quality, with clear descriptions and labels. The raw data related to the manuscript has been provided and shared. In general, the language of the manuscript is basically clear and fluent, but some sentences in this manuscript still need to be further improved in language.

Experimental design

The experimental design and result analysis are reasonable.

Validity of the findings

The result analysis is basically reasonable.

Additional comments

This manuscript reports the identification and analysis of the OSCA gene family in sunflower. The experimental design and result analysis are reasonable. However, the grammar and format of the manuscript still need to be improved.

(1) In the introduction section, the manuscript should emphasize the conservative domain characteristics of OSCA gene family, so as to provide background knowledge for the subsequent analysis.

(2) Using different Pfam IDs will identify different gene families. In line 104, ".....Hidden Markov model (PF03514) and .... ", PF03514 should be the Pfam ID for GRAS gene family, but not the Pfam ID for OSCA gene family. Therefore, the Pfam ID is incorrect. Please check it carefully. If this study really used this ID (PF03514) to identify gene families, the gene families identified in this study will no longer be OSCA gene families, but become other gene families.

(3) In the results section, some inferences seem arbitrary and lack some scientific evidences. Please revise them carefully. For example, "... indicates that the OSCA gene may be more important for functional differentiation in sunflowers", According to the fact that there is no OSCA gene of sunflower in group 4, it is lack of scientific basis to infer that OSCA gene may be more important for functional differentiation of sunflower. Another example, in line 177-180, the author infers that "this may have a certain relationship with the planting environment of different crops", which seems to lack sufficient basis. In this manuscript, there are many similar inferences that lack scientific basis. Please revise them carefully.

(4) The analysis results of cis-acting elements and WGCNA are too simple to see more analysis results.

(5) Which RNA-seq data sets were used to perform the analysis of gene expression and WGCNA? Please provide the ID of the RNA-seq data sets. The reference control are not provided in qRT-PCR.

(6) Some sentences in this manuscript still need to be further improved in language, or even rewritten.

In addition, there are many minor problems in this manuscript as following:

----Gene name should be italic and protein name should not be italic. There are so many mistakes, please check them.

----The Latin name "Helianthus annuus L" of the sunflower in the title and line 47 and "Glycine max (Linn.) Merr" in line 73 are incorrect in writing. Please check the full text.

---- In line 217, the title of " Phylogenetic tree, gene structure and motif analysis of the OSCA gene of Compositae" should remove " Phylogenetic tree ".

---- Pay attention to the use of singular and plural verbs.

----In the manuscript, there are many mistakes in the use of uppercase/lowercase and word Spelling. For example, “Gene structural” should be “Gene structure” in figure 6, “plant material” in line 93 should be "Plant material", “the open reading frame (OFR)” in line 160 should be “the open reading frame (ORF)”, "learning function.." in line 236, etc. Please carefully check these errors in the manuscript and correct them.

---- Abbreviations should have full names when they first appear.

---- The chromosome names are overlap in figure 3 and 5.

There are still many other similar errors in the manuscript. Please check and correct them.

---

## Round 0.2 · accepted · Accept

Dear Authors,

Thanks for improving the MS based on the inputs from the reviewers. Wish you all the best.

Thanks and Regards

The Section Editor mentions:
> I think that the genus/species of sunflower should be mentioned in the abstract, since it is the subject of the study.

Reviewer 2 ·

Basic reporting

Authors have revised manuscript adequately. I am satisfied with work done.

Experimental design

This is alright.

Validity of the findings

These are ok.